# Synergy and remarkable specificity of antimicrobial peptides in vivo using a systematic knockout approach

Mark Austin Hanson[1]*, Anna Dostálová[1], Camilla Ceroni[1], Mickael Poidevin[2], Shu Kondo[3], Bruno Lemaitre[1]*

[1]Global Health Institute, School of Life Science, École Polytechnique Fédérale de Lausanne (EPFL), Lausanne, Switzerland; [2]Institute for Integrative Biology of the Cell (I2BC), Université Paris-Saclay, CEA, CNRS, Université Paris Sud, Gif-sur-Yvette, France; [3]Invertebrate Genetics Laboratory, Genetic Strains Research Center, National Institute of Genetics, Mishima, Japan

**Abstract** Antimicrobial peptides (AMPs) are host-encoded antibiotics that combat invading microorganisms. These short, cationic peptides have been implicated in many biological processes, primarily involving innate immunity. In vitro studies have shown AMPs kill bacteria and fungi at physiological concentrations, but little validation has been done in vivo. We utilized CRISPR gene editing to delete most known immune-inducible AMPs of *Drosophila*, namely: 4 Attacins, 2 Diptericins, Drosocin, Drosomycin, Metchnikowin and Defensin. Using individual and multiple knockouts, including flies lacking these ten AMP genes, we characterize the in vivo function of individual and groups of AMPs against diverse bacterial and fungal pathogens. We found that *Drosophila* AMPs act primarily against Gram-negative bacteria and fungi, contributing either additively or synergistically. We also describe remarkable specificity wherein certain AMPs contribute the bulk of microbicidal activity against specific pathogens, providing functional demonstrations of highly specific AMP-pathogen interactions in an in vivo setting.
DOI: https://doi.org/10.7554/eLife.44341.001

*For correspondence:
mark.hanson@epfl.ch (MAH);
bruno.lemaitre@epfl.ch (BL)

## Introduction

While innate immune mechanisms were neglected during the decades where adaptive immunity captured most of the attention, they have become central to our understanding of immunology. Recent emphasis on innate immunity has, however, mostly focused on the first two phases of the immune response: microbial recognition and associated downstream signaling pathways. In contrast, how innate immune effectors individually or collectively contribute to host resistance has not been investigated to the same extent. The existence of multiple effectors that redundantly contribute to host resistance has hampered their functional characterization by genetic approaches (*Lemaitre and Hoffmann, 2007*). The single mutation methodology that still prevails today has obvious limits in the study of immune effectors, which often belong to large gene families. As such, our current understanding of the logic underlying the roles of immune effectors is only poorly defined. As a consequence, the key parameters that influence host survival associated with a successful immune response are not well characterized. In this paper, we harnessed the power of the CRISPR gene editing approach to study the function of *Drosophila* antimicrobial peptides in host defence both individually and collectively.

Antimicrobial peptides (AMPs) are small, cationic, usually amphipathic peptides that contribute to innate immune defence in plants and animals (*Imler and Bulet, 2005*; *Guaní-Guerra et al., 2010*; *Rolff and Schmid-Hempel, 2016*). They display potent antimicrobial activity in vitro by disrupting

**eLife digest** All animals – from humans to mice, jellyfish to fruit flies – are armed with an immune system to defend against infections. The immune system's first line of defence often involves a group of short proteins called antimicrobial peptides. These proteins are found anywhere that germs and microbes come into contact with the body, including the skin, eyes and lungs. In many cases, it is unclear how individual antimicrobial peptides work. For example, which germs are they most effective against? Do they work alone, or in a mixture of other antimicrobial peptides?

To learn more about a protein, scientists can often delete the gene that encodes it and observe what happens. Antimicrobial peptides, however, are small proteins encoded by a large number of very short genes, which makes them difficult to target with most genetic tools. Fortunately, gene editing via the CRISPR/Cas9 system can overcome many of the limitations of more traditional methods; this allowed Hanson et al. to systematically remove the antimicrobial peptide genes from fruit flies to explore how these proteins work.

In the experiments, ten antimicrobial peptide genes known from fruit flies were removed, and the flies were then infected with a variety of bacteria and fungi. Hanson et al. found that the antimicrobial peptides were effective against many bacteria, but unexpectedly they were far more important for controlling one general kind of bacterial infection, but not another kind. Further experiments showed that some of these proteins work alone, targeting only a particular species of microbe. This finding suggested that animals might fight infections by very specific bacteria with a very specific antimicrobial peptide rather than with a mixture.

By understanding how antimicrobial peptides work in more detail, scientists can learn what types of microbes they are most effective against. In the future, this information may eventually lead to the development of new types of antibiotics and better management of diseases that affect important insects, like bumblebees.

DOI: https://doi.org/10.7554/eLife.44341.002

negatively charged microbial membranes, but AMPs can also target specific microbial processes (*Park et al., 1998*; *Kragol et al., 2001*; *Rahnamaeian et al., 2015*). Their expression is induced to very high levels upon challenge to provide microbicidal concentrations in the µM range. Numerous studies have revealed unique roles that AMPs may play in host physiology including anti-tumor activity (*Suttmann et al., 2008*; *Kuroda et al., 2015*; *Araki et al., 2018*; *Parvy et al., 2019*), inflammation in aging (*Cao et al., 2013*; *Kounatidis et al., 2017*; *E et al., 2018*), involvement in memory (*Bozler et al., 2017*; *Barajas-Azpeleta et al., 2018*), mammalian immune signaling (*van Wetering et al., 2002*; *Tjabringa et al., 2003*), wound-healing (*Tokumaru et al., 2005*; *Chung et al., 2017*), regulation of the host microbiota (*Login et al., 2011*; *Mergaert et al., 2017*), tolerance to oxidative stress (*Zhao et al., 2011*; *Zheng et al., 2007*), and of course microbicidal activity (*Imler and Bulet, 2005*; *Wimley, 2010*). The fact that AMP genes are immune inducible and expressed at high levels has led to the common assumption they play a vital role in the innate immune response. However, little is known in most cases about how AMPs individually or collectively contribute to animal host defence. In vivo functional analysis of AMPs has been hampered by the sheer number and small size of these genes, making them difficult to mutate with traditional genetic tools (but e.g. see *Hoeckendorf et al., 2012*).

Since the first animal AMPs were discovered in silk moths (*Steiner et al., 1981*), insects and particularly *Drosophila melanogaster* have emerged as a powerful model for characterizing their function. There are currently seven well-characterized families of inducible AMPs in *D. melanogaster,* but we note that many genes encoding small peptides are strongly upregulated upon infection and are awaiting description (*De Gregorio et al., 2002*). The activities of the seven known AMP families of *Drosophila* have been determined either in vitro by using peptides directly purified from flies or produced in heterologous systems, or deduced by comparison with homologous peptides isolated in other insect species: Drosomycin and Metchnikowin show antifungal activity (*Fehlbaum et al., 1994*; *Levashina et al., 1995*); Cecropins (four inducible genes) and Defensin have both antibacterial and some antifungal activities (*Hultmark et al., 1980*; *Ekengren and Hultmark, 1999*; *Cociancich et al., 1993*; *Tzou et al., 2002*); and Drosocin, Attacins (four genes) and Diptericins (two genes) primarily

exhibit antibacterial activity (*Kragol et al., 2001*; *Asling et al., 1995*; *Cudic et al., 1999*; *Hedengren et al., 2000*; *Bulet et al., 1996*). In *Drosophila*, these AMPs are produced either locally at various surface epithelia in contact with environmental microbes (*Tzou et al., 2000*; *Basset et al., 2000*; *Gendrin et al., 2009*), or secreted systemically into the hemolymph, the insect blood. During systemic infection, these 14 antimicrobial peptides are strongly induced in the fat body, an organ analogous to the mammalian liver.

The systemic production of AMPs is regulated at the transcriptional level by two NF-κB pathways, the Toll and Imd pathways, which are activated by different classes of microbes. The Toll pathway is predominantly responsive to Gram-positive bacteria and fungi, and accordingly plays a major role in defence against these microbes. In contrast, the Imd pathway is activated by Gram-negative bacteria and a subset of Gram-positive bacteria with DAP-type peptidoglycan, and mutations affecting this pathway cause profound susceptibility to Gram-negative bacteria (*De Gregorio et al., 2002*; *Lemaitre et al., 1997*). However, the expression pattern of AMP genes is complex as each gene is expressed with different kinetics and can often receive transcriptional input from both pathways (*De Gregorio et al., 2002*; *Leulier et al., 2000*). This ranges from *Diptericin*, which is tightly regulated by the Imd pathway, to *Drosomycin*, whose expression is mostly regulated by the Toll pathway (*Lemaitre et al., 1997*), except at surface epithelia where *Drosomycin* is under the control of Imd signaling (*Ferrandon et al., 1998*; *Tzou et al., 2000*). While a critical role of AMPs in *Drosophila* host defence is supported by transgenic flies overexpressing a single AMP (*Tzou et al., 2002*), the specific contributions of each of these AMPs has not been tested. Indeed loss-of-function mutants for most AMP genes were not previously available due to their small size, making them difficult to mutate before the advent of CRISPR/Cas9 technology. Despite this, the great susceptibility to infection of mutants with defective Toll and Imd pathways is commonly attributed to the loss of the AMPs they regulate, though these pathways control hundreds of genes awaiting characterization (*De Gregorio et al., 2002*). Strikingly, *Clemmons et al. (2015)* recently reported that flies lacking a set of uncharacterized Toll-responsive peptides (named Bomanins) succumb to infection by Gram-positive bacteria and fungi at rates similar to *Toll*-deficient mutants (*Clemmons et al., 2015*). This provocatively suggests that Bomanins, and not AMPs, might be the predominant effectors downstream of the Toll pathway; yet synthesized Bomanins do not display antimicrobial activity in vitro (*Lindsay et al., 2018*). Thus, while today the fly represents one of the best-characterized animal immune systems, the contribution of AMPs as immune effectors is poorly defined as we still do not understand why Toll and Imd pathway mutants succumb to infection.

In this paper, we took advantage of recent gene editing technologies to delete most of the known immune inducible AMP genes of *Drosophila*. Using single and multiple knockouts, as well as a variety of bacterial and fungal pathogens, we have characterized the in vivo function of individual and groups of antimicrobial peptides. We reveal that AMPs can play highly specific roles in defence, being vital for surviving certain infections yet dispensable against others. We highlight key interactions amongst immune effectors and pathogens and reveal to what extent these defence peptides act in concert or alone.

## Results

### Generation and characterization of AMP mutants

We generated null mutants for 10 of the 14 known *Drosophila* antimicrobial peptide genes that are induced upon systemic infection. These include five single gene mutations affecting *Defensin* (*Def^{SK3}*), *Attacin C* (*AttC^{Mi}*), *Metchnikowin* (*Mtk^{R1}*), *Attacin D* (*AttD^{SK1}*) and *Drosomycin* (*Drs^{R1}*), respectively, and two small deletions removing both *Diptericins* *DptA* and *DptB* (*Dpt^{SK1}*), or the gene cluster containing *Drosocin*, and *Attacins AttA* and *AttB* (*Dro-AttAB^{SK2}*). The function of Cecropins were not assessed in this manuscript. All mutations/deletions were made using the CRISPR editing approach with the exception of *Attacin C*, which was disrupted by insertion of a *Minos* transposable element (*Bellen et al., 2011*), and the *Drosomycin* and *Metchnikowin* deletions generated by homologous recombination (*Figure 1A* and *Figure 1—figure supplement 1*). To disentangle the role of *Drosocin* and *AttA/AttB* in the *Dro-AttAB^{SK2}* deletion, we also generated an individual *Drosocin* mutant (*Dro^{SK4}*); for complete information, see *Figure 1—figure*

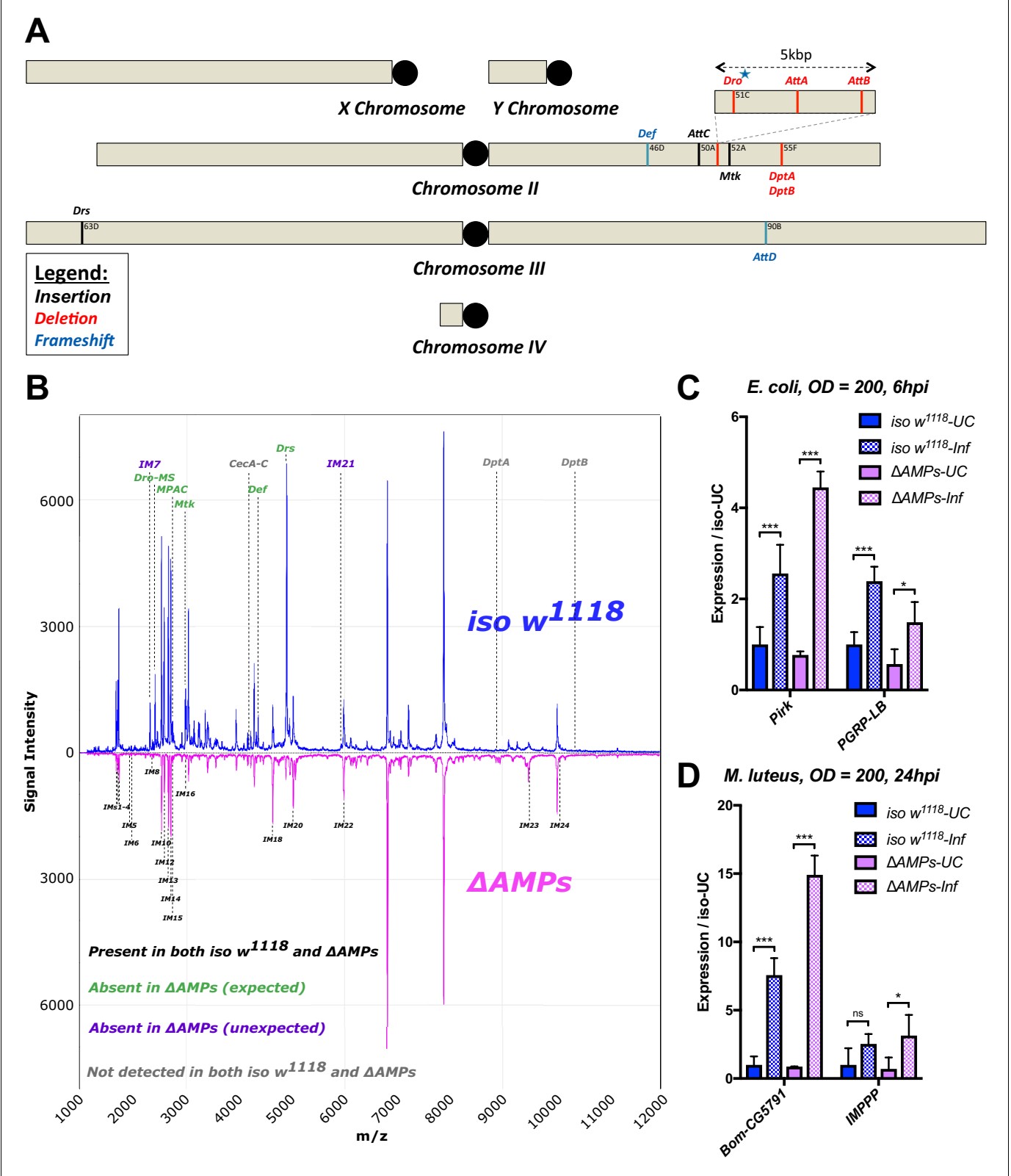

**Figure 1.** Description of *AMP* mutants. (**A**) Chromosomal locations of *AMP* genes that were deleted. Each mutation is color-coded with the mutagenic agent: black, a *Minos* insertion or homologous recombination, red, CRISPR-CAS9-mediated deletion, and blue CRISPR CAS9 mediated indel causing a nonsense peptide. (**B**) A representative MALDI-TOF analysis of hemolymph samples from immune-challenged (1:1 *E. coli* and *M. luteus* at OD600 = 200) *iso w*[1118] and *ΔAMPs* flies as described in *Uttenweiler-Joseph et al. (1998)*. No AMP-derived products were detected in the hemolymph samples of

*Figure 1 continued*

ΔAMPs flies. No signals for IM7, nor IM21 were observed in the hemolymph samples of ΔAMPs mutants suggesting that these uncharacterized immune-induced molecules are the products of AMP genes. The Imd pathway (C) and Toll pathway (D) are functional and respond to immune challenge in ΔAMPs flies. We used alternate readouts to monitor the Toll and Imd pathways: *pirk* and *PGRP-LB* for Imd pathway and *CG5791 (Bomanin)* and *IMPPP* for Toll signaling (*De Gregorio et al., 2002*; *Hanson et al., 2016*). UC = unchallenged, Inf = infected. hpi = hours post-infection. Expression normalized with *iso w$^{1118}$-UC* set to a value of 1.

DOI: https://doi.org/10.7554/eLife.44341.003

The following figure supplements are available for figure 1:

**Figure supplement 1.** Genetic description of mutations generated in this study.
DOI: https://doi.org/10.7554/eLife.44341.004

**Figure supplement 2.** ΔAMPs flies have otherwise wild-type immune reactions.
DOI: https://doi.org/10.7554/eLife.44341.005

*supplement 1*. We then isogenized these mutations for at least seven generations into the *w$^{1118}$* DrosDel isogenic genetic background (*Ryder et al., 2004*) (*iso w$^{1118}$*). Then, we recombined these seven independent mutations into a background lacking these 10 inducible AMPs referred to as 'ΔAMPs.' ΔAMPs flies were viable and showed no morphological defects. To confirm the absence of AMPs in our ΔAMPs background, we performed a MALDI-TOF analysis of hemolymph from both unchallenged and immune-challenged flies infected by a mixture of *Escherichia coli* and *Micrococcus luteus*. This analysis revealed the presence of peaks induced upon challenge corresponding to AMPs in wild-type but not ΔAMPs flies. Importantly, it also confirmed that induction of most other immune-induced molecules (IMs) (*Uttenweiler-Joseph et al., 1998*), was unaffected in ΔAMPs flies (*Figure 1B*). Of note, we failed to observe two IMs, IM7 and IM21, in our ΔAMPs flies, suggesting that these unknown peptides are secondary products of AMP genes. We further confirmed that Toll and Imd NF-κB signaling pathways were intact in ΔAMPs flies by measuring the expression of target genes of these pathways (*Figure 1C–D*). This demonstrates that *Drosophila* AMPs are not signaling molecules required for Toll or Imd pathway activity. We also assessed the role of AMPs in the melanization response, wound clotting, and hemocyte populations. After clean injury, ΔAMPs flies survive as wild-type (*Figure 1—figure supplement 2A*). We found no defect in melanization ($\chi^2$, p=0.34, *Figure 1—figure supplement 2B*) as both adults and larvae strongly melanize the cuticle following clean injury (*Figure 1—figure supplement 2C*). Furthermore, we visualized the rapid formation of clot fibers ex vivo using the hanging drop assay and PNA staining (*Scherfer et al., 2004*) in hemolymph of both wild-type and ΔAMPs larvae (*Figure 1—figure supplement 2D*). Hemocyte counting (i.e. crystal cells, FACS) did not reveal any deficiency in hemocyte populations of ΔAMPs larvae (*Figure 1—figure supplement 2E,F*, and not shown). Altogether, our study suggests that *Drosophila* AMPs are primarily immune effectors, and not regulators of innate immunity.

## AMPs are essential for combating Gram-negative bacterial infection

We used these ΔAMPs flies to explore the role that AMPs play in defence against pathogens during systemic infection. We first focused our attention on Gram-negative bacterial infections, which are combatted by Imd pathway-mediated defence in *Drosophila* (*Lemaitre and Hoffmann, 2007*). We challenged wild-type and ΔAMPs flies with six different Gram-negative bacterial species, using inoculation doses (given as OD600) selected such that at least some wild-type flies were killed. In our survival experiments, we also include Oregon R (*OR-R*) as an alternate wild-type for comparison, and *Relish* mutants (*Rel$^{E20}$*) that lack a functional Imd response and are known to be very susceptible to this class of bacteria (*Hedengren et al., 1999*) (*Figure 2*). Globally, ΔAMPs flies were extremely susceptible to all Gram-negative pathogens tested (*Figure 2*, light blue plots). The susceptibility of AMP-deficient flies to Gram-negative bacteria largely mirrored that of *Rel$^{E20}$* flies. For all Gram-negative infections tested, ΔAMPs flies show a higher bacterial count at 18 hr post-infection (hpi) indicating that AMPs actively inhibit bacterial growth, as expected of 'antimicrobial peptides' (*Figure 2—figure supplement 1A*). Use of GFP-expressing bacteria show that bacterial growth in ΔAMPs flies radiates from the wound site until spreading systemically (*Figure 2—figure supplement 1B,C*). Collectively, the use of AMP-deficient flies reveals that AMPs are major players in resistance to Gram-negative bacteria, and likely constitute an essential component of the Imd pathway's contribution for survival against these germs.

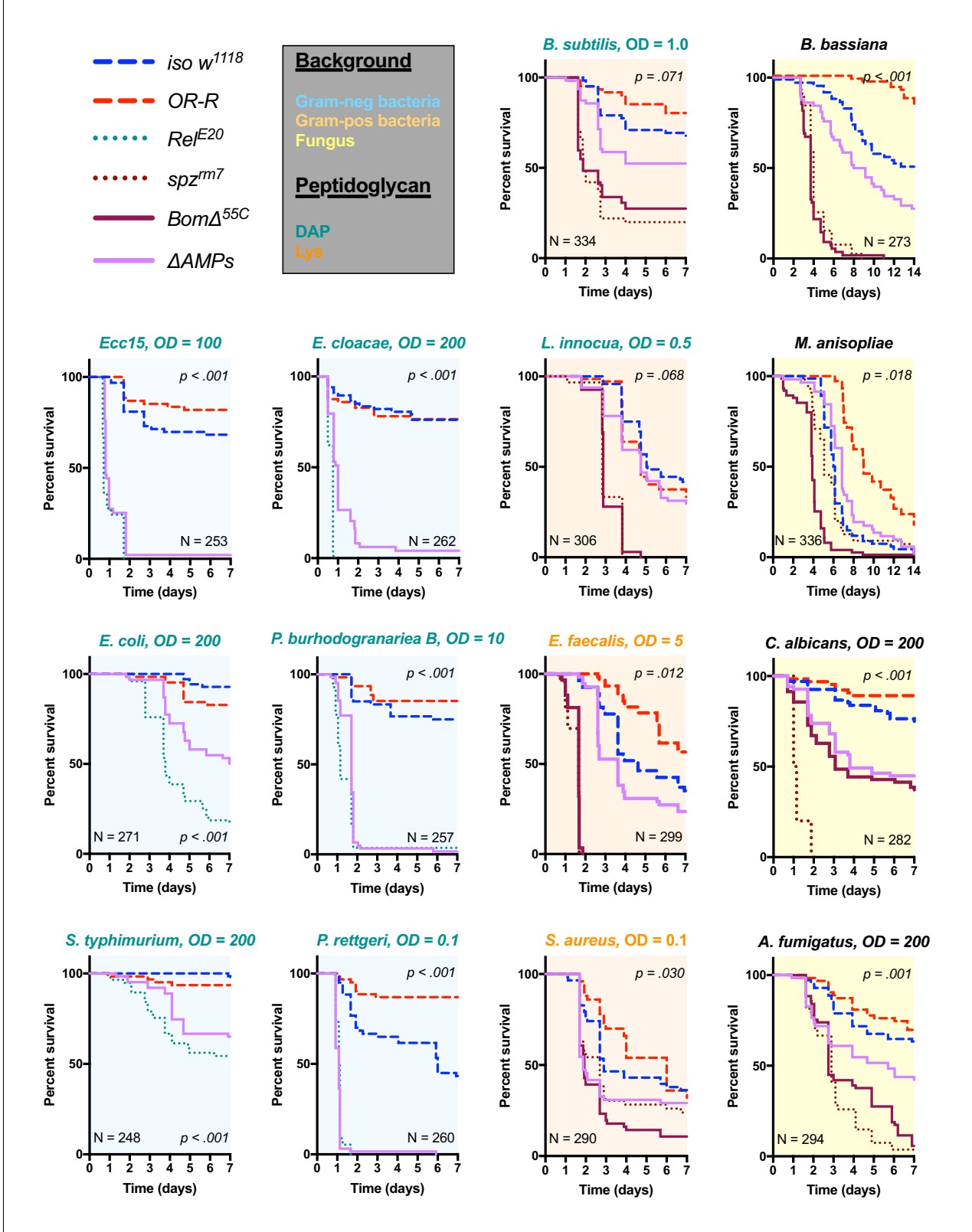

**Figure 2.** Survival of ΔAMPs flies to diverse microbial challenges. Control lines for survival experiments included two wild-types (w;Drosdel (iso w¹¹¹⁸) and Oregon R (OR-R) as an alternate wild-type), mutants for the Imd response (Rel^E20), mutants for Toll signaling (spz^rm7), and mutants for Bomanins (Bom^Δ55C). ΔAMPs flies are extremely susceptible to infection with Gram-negative bacteria (blue backgrounds). Unexpectedly, ΔAMPs flies were not markedly susceptible to infection with Gram-positive bacteria (orange backgrounds), while Bom^Δ55C flies were extremely susceptible, often mirroring

*Figure 2 continued on next page*

Figure 2 continued

$spz^{rm7}$ mutants. This pattern of $Bom^{\Delta 55C}$ susceptibility held true for fungal infections (yellow backgrounds). $\Delta AMPs$ flies are somewhat susceptible to fungal infections, but the severity shifts with different fungi. Pellet densities are reported for all systemic infections in OD at 600 nm. p-Values are given for $\Delta AMPs$ flies compared to $iso\ w^{1118}$ using a Cox-proportional hazards model. N = total number of flies in experiments. A full description of p-values relative to $iso\ w^{1118}$ can be found in **Figure 2—source data 1**.

DOI: https://doi.org/10.7554/eLife.44341.006

The following source data and figure supplement are available for figure 2:

**Source data 1.** p-Values from **Figure 2A** relative to $iso\ w^{1118}$.

DOI: https://doi.org/10.7554/eLife.44341.008

**Figure supplement 1.** $\Delta AMPs$ flies fail to suppress Gram-negative bacterial growth.

DOI: https://doi.org/10.7554/eLife.44341.007

## Bomanins and to a lesser extent AMPs contribute to resistance against Gram-positive bacteria and fungi

Previous studies have shown that resistance to Gram-positive bacteria and fungi in *Drosophila* is mostly mediated by the Toll pathway, although the Imd pathway also contributes to some extent (*Lemaitre et al., 1997*; *Leulier et al., 2000*; *Rutschmann et al., 2002*; *Tanji et al., 2007*). Moreover, a deletion removing ten uncharacterized Bomanins ($Bom^{\Delta 55C}$) induces a strong susceptibility to both Gram-positive bacteria and fungi (*Clemmons et al., 2015*), suggesting that Bomanins are major players downstream of Toll in the defence against these germs. This prompted us to explore the role of antimicrobial peptides in defence against Gram-positive bacteria and fungi. In these experiments, we additionally included *spätzle* mutant flies ($spz^{rm7}$) lacking Toll signaling as susceptible controls. We first challenged wild-type and $\Delta AMPs$ flies with two lysine-type (*E. faecalis, S. aureus*) and two DAP-type (*B. subtilis, L. innocua*) peptidoglycan-containing Gram-positive bacterial species. We observed that $\Delta AMPs$ flies display only weak or no increased susceptibility to infection with these Gram-positive bacterial species, as $\Delta AMPs$ survival rates were closer to the wild-type than to $spz^{rm7}$ mutants lacking a functional Toll pathway (**Figure 2**, orange plots), with the exception of *S. aureus*. Meanwhile, $Bom^{\Delta 55C}$ mutants consistently phenocopied $spz^{rm7}$ flies, confirming the important contribution of these peptides in defence against Gram-positive bacteria (*Clemmons et al., 2015*).

Next, we monitored the survival of $\Delta AMPs$ to the yeast *Candida albicans*, the opportunistic fungus *Aspergillus fumigatus* and two entomopathogenic fungi, *Beauveria bassiana*, and *Metarhizium anisopliae*. For the latter two, we used a natural mode of infection by spreading spores on the cuticle (*Lemaitre et al., 1997*). $\Delta AMPs$ flies were more susceptible to fungal infections with *B. bassiana*, *A. fumigatus*, and *C. albicans*, but not *M. anisopliae* (**Figure 2**, yellow plots). In all instances, $Bom^{\Delta 55C}$ mutants were as or more susceptible to fungal infection than $\Delta AMPs$ flies, approaching *Toll*-deficient mutant levels. Collectively, our data demonstrate that AMPs are major immune effectors in defence against Gram-negative bacteria and have a less essential role in defence against bacteria and fungi.

## A combinatory approach to explore AMP interactions

The impact of the $\Delta AMPs$ deletion on survival could be due to the action of certain AMPs having a specific effect, or more likely due to the combinatory action of co-expressed AMPs. Indeed, cooperation of AMPs to potentiate their microbicidal activity has been suggested by numerous in vitro approaches (*Rahnamaeian et al., 2015*; *Yu et al., 2016*; *Mohan et al., 2014*), but rarely in an in vivo context (*Zanchi et al., 2017*). Having shown that AMPs as a whole significantly contribute to fly defence, we next explored the contribution of individual peptides to this effect. To tackle this question in a systematic manner, we performed survival analyses using fly lines lacking one or several AMPs, focusing on pathogens with a range of virulence that we previously showed to be sensitive to the action of AMPs. This includes the yeast *C. albicans* and the Gram-negative bacterial species *P. burhodogranariea, P. rettgeri, Ecc15,* and *E. cloacae*. Given seven independent AMP mutations, over 100 combinations of mutants are possible, making a systematic analysis of AMP interactions a logistical nightmare. Therefore, we designed an approach that would allow us to characterize their contributions to defence by deleting groups of AMPs. To this end, we generated three groups of

combined mutants: A) flies lacking *Defensin* (Group A); *Defensin* is regulated by Imd signalling but is primarily active against Gram-positive bacteria in vitro (Imler and Bulet, 2005). B) Flies lacking three antibacterial and structurally related AMP families: the Proline-rich *Drosocin* and the Proline- and Glycine-rich *Diptericins* and *Attacins* (Group B, regulated by the Imd pathway). C) Flies lacking the two antifungal peptide genes *Metchnikowin* and *Drosomycin* (Group C, mostly regulated by the Toll pathway). We then combined these three groups to generate flies lacking AMPs from groups A and B (AB), A and C (AC), or B and C (BC). Finally, flies lacking all three groups are our Δ*AMPs* flies, which are highly susceptible to a number of infections. By screening these seven genotypes as well as individual mutants, we were able to assess potential interactions between AMPs of different groups, as well as decipher the function of individual AMPs.

## Drosomycin and metchnikowin additively contribute to defence against the yeast *C. albicans*

We first applied this AMP-groups approach to infections with the relatively avirulent yeast *C. albicans*. Previous studies have shown that Toll, but not Imd, contributes to defence against this fungus (Gottar et al., 2006; Glittenberg et al., 2011). Thus, we suspected that the two antifungal peptides, Drosomycin and Metchnikowin, could play a significant role in the susceptibility of Δ*AMPs* flies to this yeast. Consistent with this, Group C flies lacking *Metchnikowin* and *Drosomycin* were more susceptible to infection (p<0.001 relative to *iso w*[1118]) with a survival rate similar to Δ*AMPs* flies (*Figure 3A*). Curiously, AC-deficient flies that also lack *Defensin* survived better than Group C-deficient flies (Log-Rank p=0.014). We have no explanation for this interaction, but this could be due to i) a better canalization of the immune response by preventing the induction of ineffective AMPs, ii) complex biochemical interactions amongst the AMPs involved affecting either the host or pathogen or iii) differences in genetic background generated by additional recombination. We then investigated the individual contributions of *Metchnikowin* and *Drosomycin* to survival to *C. albicans*. We found that both *Mtk*[R1] and *Drs*[R1] individual mutants were somewhat susceptible to infection, but notably only *Mtk; Drs* compound mutants reached Δ*AMPs* levels of susceptibility (*Figure 3B*). This co-occurring loss of resistance appears to be primarily additive (Mutant, Cox Hazard Ratio (HR), p-value: *Mtk*[R1], HR =+1.17, p=0.008; *Drs*[R1], HR =+1.85, p<0.001; *Mtk*Drs*, HR = −0.80, p=0.116). We observed that Group C deficient flies eventually succumb to uncontrolled *C. albicans* growth by monitoring yeast titre, indicating that these AMPs indeed act by suppressing yeast growth (*Figure 3C*).

In conclusion, our study provides an in vivo validation of the potent antifungal activities of Metchnikowin and Drosomycin (Fehlbaum et al., 1994; Levashina et al., 1995), and highlights a clear example of additive cooperation of AMPs.

## AMPs synergistically contribute to defence against *P. burhodogranariea*

We next analyzed the contribution of AMPs in resistance to infection with the moderately virulent Gram-negative bacterium *P. burhodogranariea*. We found that Group B mutants lacking *Drosocin*, the two *Diptericins*, and the four *Attacins*, were as susceptible to infection as Δ*AMPs* flies (*Figure 4A*), while flies lacking the antifungal peptides Drosomycin and Metchnikowin (Toll-regulated, Group C) resisted the infection as wild-type. Flies lacking *Defensin* (Group A) showed an intermediate susceptibility, but behave as wild-type in the additional absence of Toll Group C peptides (Group AC). Thus, we again observed a better survival rate with the co-occurring loss of Group A and C peptides (see possible explanation above). In this case, Group A flies were susceptible while AC flies were not.

Following the observation that Group B flies were as susceptible as Δ*AMPs* flies, we sought to better decipher the contribution of each Group B AMP to resistance to *P. burhodogranariea*. We observed that mutants for *Drosocin* alone (*Dro*[SK4]), or the *DiptericinA/B* deficiency were not susceptible to this bacterium (*Figure 4B*). We additionally saw no marked susceptibility of *Drosocin-Attacin A/B*-deficient flies, nor *Attacin C* or *Attacin D* mutants (not shown). Interestingly, we found that compound mutants lacking *Drosocin* and *Attacins A, B, C,* and *D* (*Figure 4B*: 'Δ*Dro*, Δ*Att*'), or *Drosocin* and *Diptericins DptA* and *DptB* ('Δ*Dro*, Δ*Dpt*') displayed an intermediate susceptibility. Only the Group B mutants lacking *Drosocin*, all *Attacins*, and both *Diptericins* (Δ*Dro*, Δ*Att*, Δ*Dpt*)

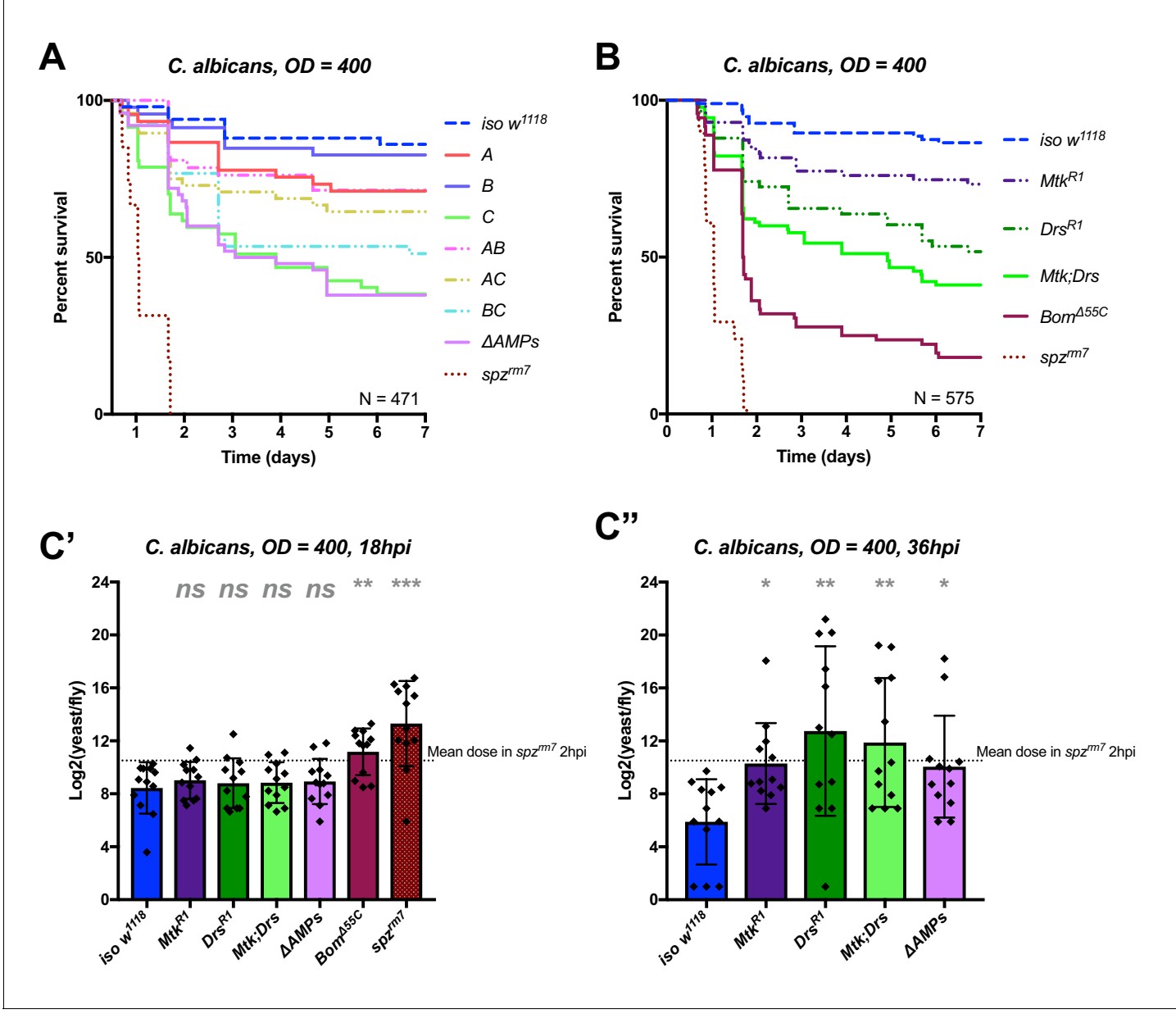

**Figure 3.** Identification of AMPs involved in the susceptibility of ΔAMPs flies to *C. albicans*. (**A**) Survival of mutants for groups of AMPs reveals that loss of only Toll-responsive Group C peptides (Metchnikowin and Drosomycin) is required to recapitulate the susceptibility of ΔAMPs flies. Co-occurring loss of groups A and C has a net protective effect (*A\*C*: HR = −1.71, p=0.002). (**B**) Further dissection of Group C mutations reveals that both Metchnikowin and Drosomycin contribute to resist *C. albicans* survival (p=0.008 and p<0.001, respectively). The interaction of Metchnikowin and Drosomycin was not different from the sum of their individual effects (*Mtk\*Drs*: HR = −0.80, p=0.116). (**C**) Fungal loads of individual flies at 18 hpi. At this time point, *Bom^{Δ55C}* mutants and *spz^{rm7}* flies have already failed to constrain *C. albicans* growth (C'). Fungal titres at 36hpi (C''), a time point closer to mortality for many AMP mutants, show that some AMP mutants fail to control fungal load, while wild-type flies consistently controlled fungal titre. One-way ANOVA: not significant = *ns*, p<0.05 = \*, p<0.01 = \*\*, and p<0.001 = \*\*\* relative to *iso w^{1118}*.

DOI: https://doi.org/10.7554/eLife.44341.009

phenocopied ΔAMPs flies (**Figure 4B**), with synergistic statistical interactions observed upon co-occurring loss of *Attacins* and *Diptericins* (*ΔAtt\*ΔDpt*: HR =+1.45, p<0.001); we emphasize here that this synergistic interaction solely reflects that the effect on survival of combining these mutations is greater than the sum effect of the individual mutations (discussed later). By 6hpi, bacterial titres of individual flies already showed significant differences in the most susceptible genotypes (**Figure 4C**), although these differences were reduced by 18 hpi likely owing to the high chronic load *P.*

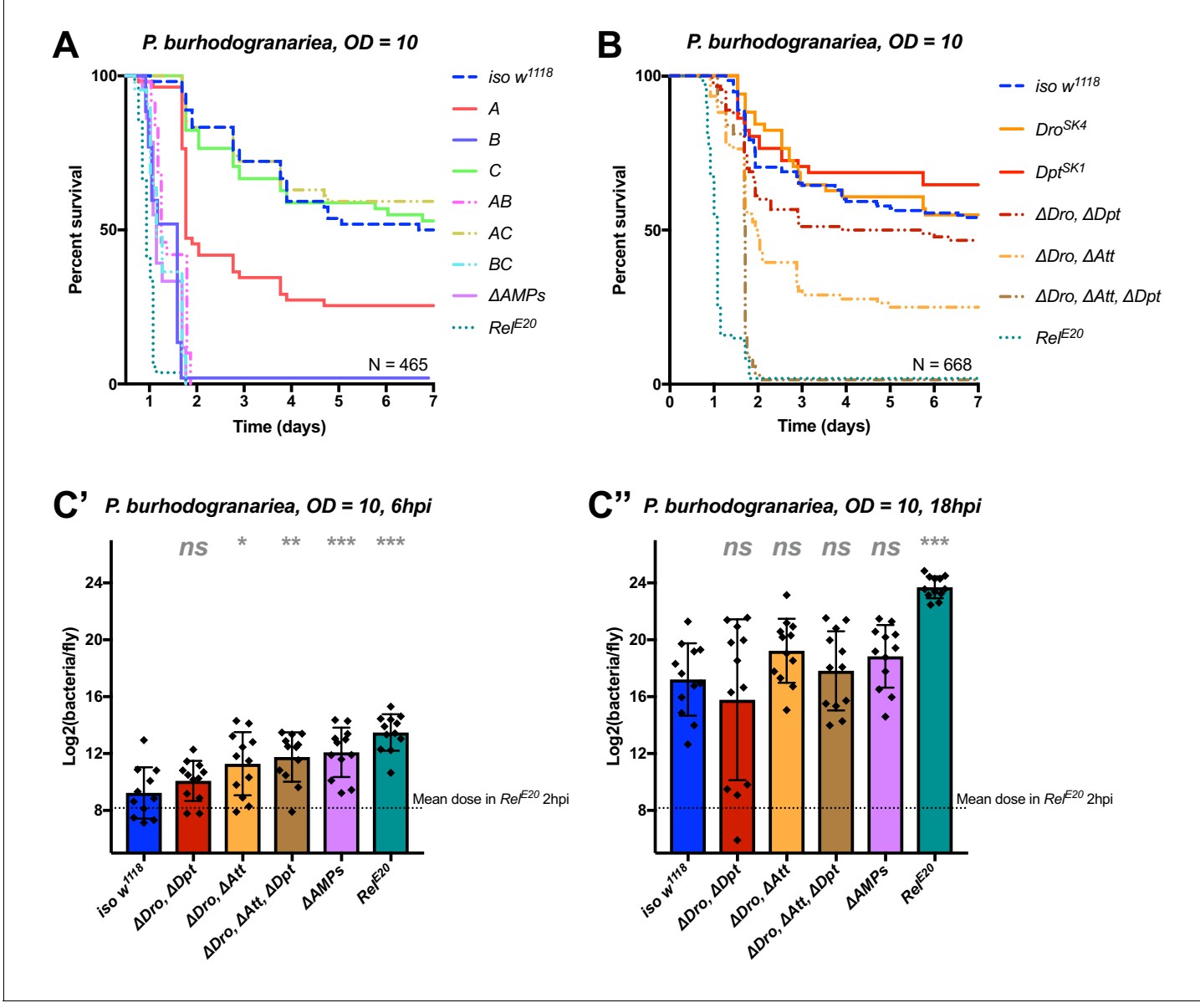

**Figure 4.** Identification of AMPs involved in the susceptibility of ΔAMPs flies to *P. burhodogranariea*. (A) Survival of mutants for groups of AMPs reveals that loss of Imd-responsive Group B peptides (Drosocin, Attacins, and Dptericins) recapitulates the susceptibility of ΔAMPs flies. Loss of the Group A peptide Defensin also resulted in strong susceptibility (p<0.001) (and see *Figure 4—figure supplement 1*). (B) Further dissection of AMPs deleted in Group B reveals that only the loss of all Drosocin, Attacin, and Dptericin gene families leads to susceptibility similar to ΔAMPs flies. Simultaneous loss of *Attacins* and *Dptericins* results in a synergistic loss of resistance (ΔAtt*ΔDpt: HR =+1.45, p<0.001). (C) Bacterial loads of individual flies at 6 hpi (C'). At this time point, most AMP mutants had significantly higher bacterial loads compared to wild-type flies. At 18 hpi (C''), differences in bacterial load are reduced, likely owing to the high chronic load *P. burhodogranariea* establishes even in surviving flies (*Duneau et al., 2017*). Meanwhile *Rel^E20* flies succumb ~18 hr earlier than ΔAMPs flies in survival experiments, and already have significantly higher loads. One-way ANOVA: not significant = *ns*, p<0.05 = *, p<0.01 = **, and p<0.001 = *** relative to *iso w^1118*.

DOI: https://doi.org/10.7554/eLife.44341.010

The following figure supplement is available for figure 4:

**Figure supplement 1.** Further dissecting effects of AMP groups.

DOI: https://doi.org/10.7554/eLife.44341.011

*burhodogranariea* establishes in surviving flies (*Duneau et al., 2017*; also see *Figure 2—figure supplement 1A*).

Collectively, the use of various compound mutants reveals that several Imd-responsive AMPs, notably Drosocin, Attacins, and Diptericins, jointly contribute to defence against *P. burhodogranariea* infection. A strong susceptibility of Group B flies was also observed upon infection with *Ecc15*, another Gram-negative bacterium commonly used to infect flies (*Neyen et al., 2014*) (*Figure 4—figure supplement 1B* ).

## *Diptericins* alone contribute to defence against *P. rettgeri*

We continued our exploration of AMP interactions using our AMP groups approach with the fairly virulent *P. rettgeri* (strain Dmel), a strain isolated from wild-caught *Drosophila* hemolymph (*Juneja and Lazzaro, 2009*). We were especially interested by this bacterium as previous studies (*Unckless et al., 2015*; *Unckless and Lazzaro, 2016*) have shown a correlation between susceptibility to *P. rettgeri* and a polymorphism in the *Diptericin A* gene pointing to a specific AMP-pathogen interaction. Use of compound mutants revealed only loss of Group B AMPs was needed to reach the susceptibility of Δ*AMPs* and *Rel*[E20] flies (*Figure 5A*). Use of individual mutant lines, however, revealed a pattern overtly different from *P. burhodogranariea*, as the sole *Diptericin A/B* deficiency caused susceptibility similar to Group B, Δ*AMPs*, and *Rel*[E20] flies (*Figure 5B,C*). We further confirmed this susceptibility using a *DptA RNAi* construct (*Figure 5—figure supplement 1A, B*). Moreover, flies carrying the *Dpt*[SK1] mutation over a deficiency (*Df(2R)Exel6067*) were also highly susceptible to *P. rettgeri* (*Figure 5D*). Interestingly, flies that were heterozygotes for *Dpt*[SK1] or the *Df(2R)Exel6067* that still have one copy of the two *Diptericins* were markedly susceptible to infection with *P. rettgeri* (*Figure 5D*). This indicates that a full transcriptional output of *Diptericin* is required over the course of the infection to resist *P. rettgeri* (*Figure 5E*). Altogether, our results suggest that only the *Diptericin* gene family, amongst the many AMPs regulated by the Imd pathway, provides the full AMP-based contribution to defence against this bacterium. To test this hypothesis, we generated a fly line lacking all the AMPs except *DptA* and *DptB* (Δ*AMPs*[+Dpt]). Strikingly, Δ*AMPs*[+Dpt] flies have the same survival rate as wild-type flies, further emphasizing the specificity of this interaction (*Figure 5B*). Bacterial counts confirm that the susceptibility of these *Diptericin* mutants arises from an inability of the host to suppress bacterial growth (*Figure 5C*).

Collectively, our study shows that *Diptericins* are critical to resist *P. rettgeri*, while they play an important but less essential role in defence against *P. burhodogranariea* infection. We were curious whether *Diptericin*'s major contribution to defence observed with *P. rettgeri* could be generalized to other members of the genus *Providencia*. An exclusive role for *Diptericins* was also found for the more virulent *P. stuartii* (*Figure 5—figure supplement 1C*), but not for other *Providencia* species tested (*P. burhodogranariea*, *P. alcalifaciens*, *P. sneebia*, *P. vermicola*) (data not shown).

## *Drosocin* is critical to resist infection with *E. cloacae*

In the course of our exploration of AMP-pathogen interactions, we identified another highly specific interaction between *E. cloacae* and Drosocin. Use of compound mutants revealed that alone, Group B flies were already susceptible to *E. cloacae*. Meanwhile, Group AB flies additionally lacking *Defensin* reached Δ*AMPs* levels of susceptibility, while Group A and Group C flies resisted as wild-type-Meanwhile, Group AB flies reached Δ*AMPs* levels of susceptibility, while Group A and Group C flies resisted as wild-type (*Figure 6A*). The high susceptibility of Group AB flies results from a synergistic statistical interaction amongst Group A (Defensin) and Group B peptides in defence against *E. cloacae* (*A\*B*, HR =+2.55, p=0.003).

We chose to further explore the AMPs deleted in Group B flies, as alone this genotype already displayed a strong susceptibility. Use of individual mutant lines revealed that mutants for *Drosocin* alone (*Dro*[SK4]) or the *Drosocin-Attacin A/B* deficiency (*Dro-AttAB*[SK2]), but not *AttC*, *AttD*, nor *Dpt*[SK1] (not shown), recapitulate the susceptibility observed in Group B flies (*Figure 6B*). At 18 hpi, both *Dro*[SK4] and Δ*AMPs* flies had significantly higher bacterial loads compared to wild-type flies, while *Rel*[E20] mutants were already moribund with much higher bacterial loads (*Figure 6C*). Indeed, the deletion of *Drosocin* alone alters the fly's ability to control the otherwise avirulent *E. cloacae* upon inoculations using OD = 200 (~39,000 bacteria, *Figure 6A–C*) or even OD = 10 (~7000 bacteria, *Figure 6—figure supplement 1A*).

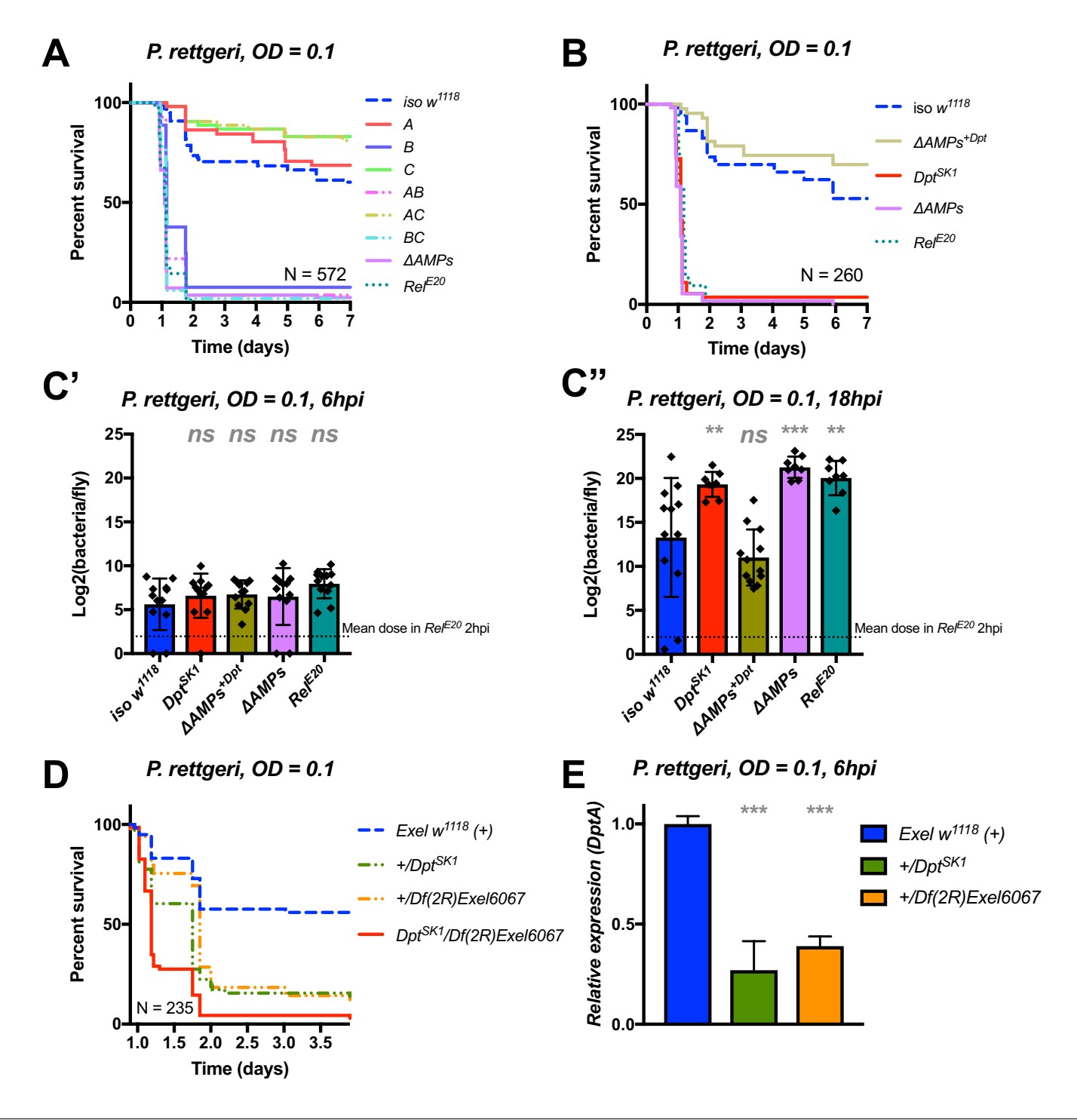

**Figure 5.** Identification of AMPs involved in the susceptibility of ΔAMPs flies to *P. rettgeri*. (**A**) Survival of mutants for groups of AMPs reveals that only loss of Imd-responsive Group B peptides (Drosocin, Attacins, and Diptericins) recapitulates the susceptibility of ΔAMPs flies. (**B**) Further dissection of the mutations affected in Group B reveals that only the loss of Diptericins (*Dpt^SK1^*) leads to susceptibility similar to ΔAMPs flies. Remarkably, flies lacking all other AMPs (ΔAMPs^+Dpt^) resist as wild-type. (**C**) Bacterial loads of individual flies are similar at 6hpi (C'), but by 18hpi (C''), *Dpt* mutants and *Rel^E20^* flies have all failed to control *P. rettgeri* growth. (**D**) Heterozygote flies for *Dpt^SK1^* and a deficiency including the *Diptericins* and flanking genes (*Df(2R) Exel6067*) recapitulates the susceptibility of *Diptericin* mutants. Intriguingly, heterozygotes with one functional copy of the Diptericins (*+/Dpt^SK1^* or *+/Df(2R)Exel6067*) are nonetheless highly susceptible to infection. (**E**) *Diptericin A* transcriptional output is strongly reduced in heterozygotes 6 hpi compared to wild-type flies. One-way ANOVA: not significant = *ns*, $p<0.05$ = *, $p<0.01$ = **, and $p<0.001$ = *** relative to *iso w^1118^*.

*Figure 5 continued on next page*

*Figure 5 continued*

DOI: https://doi.org/10.7554/eLife.44341.012

The following figure supplement is available for figure 5:

**Figure supplement 1.** Additional validation of the role of *Diptericin* in resistance to *Providencia*.

DOI: https://doi.org/10.7554/eLife.44341.013

We confirmed the high susceptibility of *Drosocin* mutant flies to *E. cloacae* in various contexts: transheterozygote flies carrying *Dro^SK4^* over a *Drosocin* deficiency (*Df(2R)BSC858*) that also lacks flanking genes including *AttA* and *AttB* ((**Figure 6D**), the *Dro ^SK4^* mutations in an alternate genetic background (*yw*, **Figure 6E**), and, *Drosocin RNAi* (**Figure 6—figure supplement 1B,C**). Thus, we recovered two highly specific AMP-pathogen interactions: Diptericins are essential to combat *P. rettgeri* infection, while *Drosocin* is paramount to surviving *E. cloacae* infection.

## Discussion

### A combinatory approach to study AMPs

Despite the recent emphasis on innate immunity, little is known on how immune effectors contribute individually or collectively to host defence, exemplified by the lack of in depth in vivo functional characterization of *Drosophila* AMPs. Taking advantage of new gene editing approaches, we developed a systematic mutation approach to study the function of *Drosophila* AMPs. With seven distinct mutations, we were able to generate a fly line lacking 10 AMPs that are known to be strongly induced during the systemic immune response. A striking first finding is that ΔAMPs flies were perfectly healthy and have an otherwise wild-type immune response. This indicates that in contrast to mammals (*van Wetering et al., 2002*), these *Drosophila* AMPs are not likely to function as signaling molecules. Using a systemic mode of infection that induces AMP expression in the fat body and hemocytes, we found that most flies lacking a single AMP family exhibited a higher susceptibility to certain pathogens consistent with their in vitro activity. We found activity of Diptericins against *P. rettgeri*, Drosocin against *E. cloacae*, Drosomycin and Metchnikowin against *C. albicans*, and Defensin against *P. burhodogranariea*. In most cases, the susceptibility of single mutants was slight, and the contribution of individual AMPs could be revealed only when combined to other AMP mutations as illustrated by the susceptibility of *Drosocin, Attacin,* and *Diptericin* combined mutants to *P. burhodogranariea*. Thus, the use of compound rather than single mutations provides a better strategy to decipher the contribution of AMPs to host defence. Our findings are consistent with a previous study using flies that constitutively expressed individual peptides (*Tzou et al., 2002*), which showed an activity of Drosomycin against *A. fumigatus* and Attacin against *Ecc15*. Beyond the systemic immune response, AMPs are also expressed in many tissues such as the gut and trachea (*Ferrandon et al., 1998*; *Tzou et al., 2000*). Future studies should investigate the role of AMPs in these local epithelial immune responses.

### AMPs and Bomanins are essential contributors to Toll and Imd pathway mediated host defence

The Toll and Imd pathways provide a paradigm of innate immunity, illustrating how two distinct pathways link pathogen recognition to distinct but overlapping sets of downstream immune effectors (*Lemaitre and Hoffmann, 2007*; *Buchon et al., 2014*). However, a method of deciphering the contributions of the different downstream effectors to the specificity of these pathways remained out of reach, as mutations in these immune effectors were lacking. Our study shows that AMPs contribute greatly to resistance to Gram-negative bacteria. Consistent with this, ΔAMPs flies are almost as susceptible as Imd-deficient mutants to most Gram-negative bacteria. In contrast, flies lacking AMPs were only slightly more susceptible to Gram-positive bacteria and fungal infections compared to wild-type flies, and this susceptibility rarely approached the susceptibility of *Bomanin* mutants. It is possible that additional loss of *Cecropins* would further increase the sensitivity of ΔAMPs flies to bacteria or fungi. This may be due to the cell walls of Gram-negative bacteria being thinner and more fluid than the rigid cell walls of Gram-positive bacteria (*Fayaz et al., 2010*), consequently making Gram-negative bacteria more prone to the action of pore-forming cationic peptides. It would be

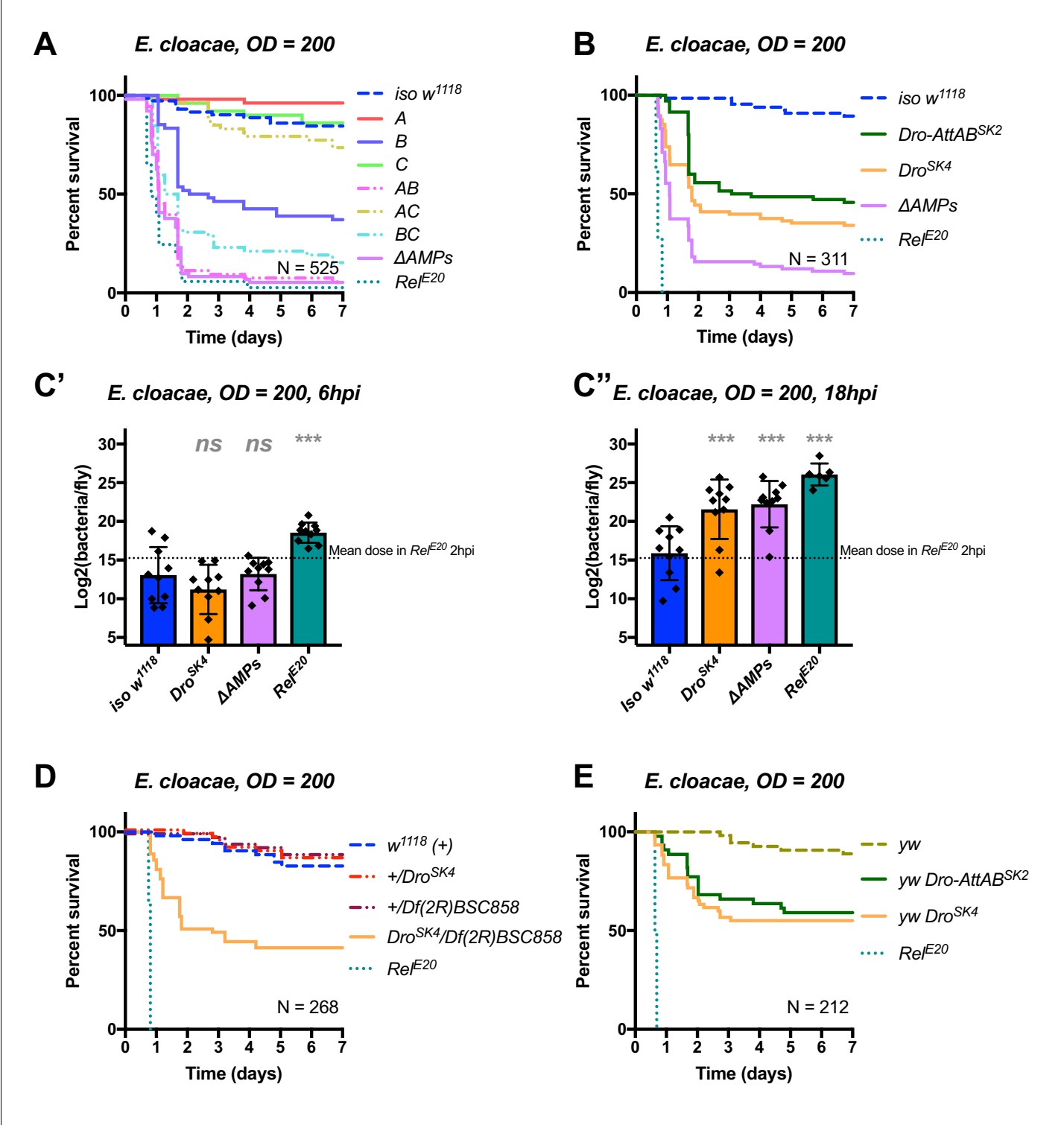

**Figure 6.** Identification of AMPs involved in the susceptibility of *ΔAMPs* flies to *E.cloacae*. (**A**) Survival of mutants for groups of AMPs reveals that loss of Imd-responsive Group B peptides (Drosocin, Attacins, and Diptericins) results in a strong susceptibility to infection (p<0.001), while loss of Group A or C peptides alone resists as wild-type (p>0.1 each). Group AB flies were as susceptible as *ΔAMPs* flies, and we observed a synergistic interaction between Group A and B mutations (A*B: HR =+2.55, p=0.003). (**B**) Further dissection of the mutations in Group B revealed that loss of *Drosocin* alone (*Dro^SK4^*), or a deficiency lacking both *Drosocin* and *Attacins AttA* and *AttB* (*Dro-AttAB^SK2^*) recapitulates the susceptibility of Group B flies. (**C**) By 18hpi, bacterial loads in individual *Drosocin* mutants or *Rel^E20^* flies are significantly higher than wild-type. (**D**) Heterozygote flies for *Dro^SK4^* and *Df(2R)BSC858* (a

*Figure 6 continued on next page*

*Figure 6 continued*

deficiency removing *Drosocin, Attacins AttA and AttB*, and other genes) are strongly susceptible to *E. cloacae* infection. (E) *Drosocin* mutants in an alternate genetic background (*yw*) are susceptible to *E. cloacae*. One-way ANOVA: not significant = *ns*, and $p<0.001$ = *** relative to *iso w$^{1118}$*.

DOI: https://doi.org/10.7554/eLife.44341.014

The following figure supplement is available for figure 6:

**Figure supplement 1.** Additional validation of the role of *Drosocin* in defence against *E. cloacae*.

DOI: https://doi.org/10.7554/eLife.44341.015

interesting to know if the specificity of AMPs to primarily combatting Gram-negative bacteria is also true in other species.

Based on our study and *Clemmons et al. (2015)*, we can now explain the susceptibility of Toll and Imd mutants at the level of the effectors, as we show that mutations affecting Imd-pathway responsive antibacterial peptide genes are highly susceptible to Gram-negative bacteria while the Toll-responsive targets Drosomycin, Metchnikowin, and especially the Bomanins, confer resistance to fungi and Gram-positive bacteria. Thus, the susceptibility of these two pathways to different sets of microbes not only reflects specificity at the level of recognition, but can now also be translated to the activities of downstream effectors. It remains to be seen how Bomanins contribute to the micro-bicidal activity of immune-induced hemolymph, as attempts to synthesize Bomanins have not revealed direct antimicrobial activity (*Lindsay et al., 2018*). It should also be noted that many putative effectors downstream of Toll and Imd remain uncharacterized, and so could also contribute to host defence beyond these AMPs and Bomanins.

## AMPs act additively and synergistically to suppress bacterial growth in vivo

In the last few years, numerous in vitro studies have focused on the potential for synergistic interactions of AMPs in microbial killing (*Rahnamaeian et al., 2015*; *Yu et al., 2016*; *Zanchi et al., 2017*; *Yan and Hancock, 2001*; *Nuding et al., 2014*; *Zerweck et al., 2017*; *Chen et al., 2005*; *Stewart et al., 2014*; *Zdybicka-Barabas et al., 2012*). Our collection of AMP mutant fly lines placed us in an ideal position to investigate AMP interactions in an in vivo setting. While Toll-responsive AMPs (Group C: Metchnikowin, Drosomycin) additively contributed to defence against the yeast *C. albicans*, we found that certain combinations of AMPs have synergistic contributions to defence against *P. burhodogranariea*. Synergistic loss of resistance may arise in two general fashions: first, co-operation of AMPs using similar mechanisms of action may breach a threshold microbicidal activity whereupon pathogens are no longer able to resist. This may be the case for the synergistic effect of Diptericins and Attacins against *P. burhodogranariea*, as only co-occurring loss of both these related glycine-rich peptide families (*Hedengren et al., 2000*) led to complete loss of resistance. Alternatively, synergy may arise due to complementary mechanisms of action, whereupon one AMP potentiates the other AMP's ability to act. For instance, the action of the bumblebee AMP Abaecin, which binds to the molecular chaperone DnaK to inhibit bacterial DNA replication, is potentiated by the presence of the pore-forming peptide Hymenoptaecin (*Rahnamaeian et al., 2016*). *Drosophila* Drosocin is highly similar to Abaecin and the related peptide Apidecin, including O-gly-cosylation of a critical threonine residue (*Imler and Bulet, 2005*; *Hanson et al., 2016*), and thus likely acts in a similar fashion. Furthermore, *Drosophila* Attacin C is matured into both a glycine-rich peptide and a Drosocin-like peptide called MPAC (*Rabel et al., 2004*). As such, co-occuring loss of Drosocin, MPAC, and other possible MPAC-like peptides encoded by the Attacin/Diptericin superfamily may be responsible for the synergistic loss of resistance in *Drosocin, Attacin, Diptericin* combined mutants.

## AMPs can act with great specificity against certain pathogens

It is commonly thought that the innate immune response lacks the specificity of the adaptive immune system, which mounts directed defences against specific pathogens. Accordingly for innate immunity, the diversity of immune-inducible AMPs can be justified by the need for generalist and/or co-operative mechanisms of microbial killing. However, an alternate explanation may be that innate immunity expresses diverse AMPs in an attempt to hit the pathogen with a 'silver bullet:' an AMP

specifically attuned to defend against that pathogen. Here, we provide a demonstration in an in vivo setting that such a strategy may actually be employed by the innate immune system. Remarkably, we recovered not just one, but two examples of exquisite specificity in our laborious but relatively limited assays.

*Diptericin* has previously been highlighted for its important role in defence against *P. rettgeri* (*Unckless et al., 2016*), but it was previously unknown whether other AMPs may confer defence in this infection model. Astoundingly, flies mutant for the other inducible AMPs resisted *P. rettgeri* infection as wild-type, while only *Diptericin* mutants succumbed to infection. This means that Diptericins do not co-operate with these other AMPs in defence against *P. rettgeri* and are solely responsible for defence in this specific host-pathogen interaction. Moreover, +/*Dpt$^{SK1}$* heterozygote flies were nonetheless extremely susceptible to infection, demonstrating that a full transcriptional output over the course of infection is required to effectively prevent pathogen growth. A previous study has shown that ~7 hpi appears to be the critical time point at which *P. rettgeri* either grows unimpeded or the infection is controlled (*Duneau et al., 2017*). This time point correlates with the time at which the *Diptericin* transcriptional output is in full-force (*Lemaitre et al., 1997*). Thus, a lag in the transcriptional response in *Dpt$^{SK1}$/+* flies likely prevents the host from reaching a competent Diptericin concentration, indicating that *Diptericin* expression level is a key factor in successful host defence.

We also show that *Drosocin* is specifically required for defence against *E. cloacae*. This striking finding validates previous biochemical analyses showing Drosocin in vitro activity against several Enterobacteriaceae, including *E. cloacae* (*Bulet et al., 1996*). As Δ*AMPs* flies are more susceptible than *Drosocin* single mutants, other AMPs also contribute to Drosocin-mediated control of *E. cloacae*. As highlighted above, Drosocin is similar to other Proline-rich AMPs (e.g. Abaecin, Pyrrhocoricin) that have been shown to target bacterial DnaK (*Kragol et al., 2001*; *Rahnamaeian et al., 2015*). Alone, these peptides still penetrate bacteria cell walls through their uptake by bacterial permeases (*Rahnamaeian et al., 2016*; *Narayanan et al., 2014*). Thus, while Drosocin would benefit from the presence of pore-forming toxins to enter bacterial cells (*Rahnamaeian et al., 2016*), the veritable 'stake to the heart' is likely the plunging of Drosocin itself into vital bacterial machinery.

## On the role of AMPs in host defence

It has often been questioned why flies should need so many AMPs (*Lemaitre and Hoffmann, 2007*; *Rolff and Schmid-Hempel, 2016*; *Unckless and Lazzaro, 2016*). A common idea, supported by in vitro experiments (*Rahnamaeian et al., 2015*; *Yan and Hancock, 2001*; *Zdybicka-Barabas et al., 2012*) is that AMPs work as cocktails, wherein multiple effectors are needed to kill invading pathogens. However, we find support for an alternative hypothesis that suggests AMP diversity may be due to highly specific interactions between AMPs and subsets of pathogens that they target. Burgeoning support for this idea also comes from recent evolutionary studies that show *Drosophila* and vertebrate AMPs experience positive selection (*Unckless et al., 2015*; *Unckless and Lazzaro, 2016*; *Hanson et al., 2016*; *Chapman et al., 2018*; *Hellgren and Sheldon, 2011*; *Tennessen and Blouin, 2008*; *Sackton, 2019*), a hallmark of host-pathogen evolutionary conflict. Our functional demonstrations of AMP-pathogen specificity, using naturally relevant pathogens (*Juneja and Lazzaro, 2009*; *Cox and Gilmore, 2007*), suggest that such specificity is fairly common, and that certain AMPs can act as the arbiters of life or death upon infection by certain pathogens. This stands in contrast to the classical view that the AMP response contains such redundancy that single peptides should have little effect on organism-level immunity (*Rolff and Schmid-Hempel, 2016*; *Unckless et al., 2015*; *Tzou et al., 2000*; *Unckless and Lazzaro, 2016*). Nevertheless, it seems these immune effectors play non-redundant roles in defence.

By providing a long-awaited in vivo functional validation for the role of AMPs in host defence, we also pave the way for a better understanding of the functions of immune effectors. Our approach of using multiple compound mutants, now possible with the development of new genome editing approaches, was especially effective to decipher the logic of immune effectors. Understanding the role of AMPs in innate immunity holds great promise for the development of novel antibiotics (*Chung et al., 2017*; *Mylonakis et al., 2016*; *Mahlapuu et al., 2016*), insight into autoimmune diseases (*Schluesener et al., 1993*; *Gilliet and Lande, 2008*; *Sun et al., 2015*; *Kumar et al., 2016*), and given their potential for remarkably specific interactions, perhaps in predicting key parameters that predispose individuals or populations to certain kinds of infections (*Unckless et al.,*

*2015*; *Unckless and Lazzaro, 2016*; *Chapman et al., 2018*). Finally, our set of isogenized *AMP* mutant lines provides long-awaited tools to decipher the role of AMPs not only in systemic immunity, but also in local immune responses, and the various roles that AMPs may play in aging, neurodegeneration, anti-tumor activity, regulation of the microbiota and more, where disparate evidence has pointed to their involvement.

## Materials and methods

### Drosophila genetics and mutant generation

The DrosDel (*Ryder et al., 2004*) isogenic $w^{1118}$ (iso $w^{1118}$) wild type was used as a genetic background for mutant isogenization. Alternate wild-types used throughout include Oregon R (*OR-R*), $w^{1118}$ from the Vienna Drosophila Resource Centre, and the Canton-S isogenic line Exelexis $w^{1118}$, which was kindly provided by Brian McCabe. $Bom^{\Delta 55C}$ mutants were generously provided by Steven Wasserman, and $Bom^{\Delta 55C}$ was isogenized into the iso $w^{1118}$ background. $Rel^{E20}$ and $spz^{rm7}$ iso $w^{1118}$ flies were provided by Luis Teixeira (*Hedengren et al., 1999*; *Ferreira et al., 2014*). Prophenoloxidase mutants (ΔPPO) are described in *Dudzic et al. (2015)*. P-element mediated homologous recombination according to *Baena-Lopez et al. (2013)* was used to generate mutants for *Mtk* ($Mtk^{R1}$) and *Drs* ($Drs^{R1}$). Plasmids were provided by Mickael Poidevin. *Attacin C* mutants ($AttC^{Mi}$, #25598), the *Diptericin* deficiency (*Df(2R)Exel6067*, #7549), the *Drosocin* deficiency (*Df(2R)BSC858*, #27928), *UAS-Diptericin RNAi* ($Dpt^{RNAi}$, #53923), *UAS-Drosocin RNAi* ($Dro^{RNAi}$, #67223), and *Actin5C-Gal4* (*ActGal4*, #4414) were ordered from the Bloomington stock centre (stock #s included). CRISPR mutations were performed by Shu Kondo according to Kondo and Ueda (*Kondo and Ueda, 2013*), and full descriptions are given in *Figure 1—figure supplement 1*. In brief, flies deficient for *Drosocin, Attacin A,* and *Attacin B* ($Dro$-$AttAB^{SK2}$), and *Diptericin A* and *Diptericin B* ($Dpt^{SK1}$) were produced by gene region deletion specific to those AMPs without affecting other genes. Single mutants for *Defensin* ($Def^{SK3}$), *Drosocin* ($Dro^{SK4}$), and *Attacin D* ($AttD^{SK1}$) are small indels resulting in the production of short (80–107 residues) nonsense peptides. Mutations were isogenized for a minimum of seven generations into the iso $w^{1118}$ background prior to subsequent recombination. It should be noted that Group A flies were initially thought to be a double mutant for both *Defensin* and the *Cecropin* cluster, resulting from a combination of $Def^{SK3}$ and a CRISPR-induced *Cecropin* deletion (called $Cec^{SK6}$). It was subsequently shown that $Cec^{SK6}$ is a complex aberration at the *Cecropin* locus that retains a wild-type copy of the *Cecropin* cluster. This re-arranged *Cecropin* locus does not contribute significantly to the susceptibility of Group A flies, as Group A was not different from $Def^{SK3}$ alone (Log-Rank p=0.818; *Figure 4—figure supplement 1A*). Thus, group A flies were considered as single $Def^{SK3}$ mutants.

### Microbial culture conditions

Bacteria were grown overnight on a shaking plate at 200 rpm in their respective growth media and temperature conditions, and then pelleted by centrifugation at 4°C. These bacterial pellets were diluted to the desired optical density at 600 nm (OD) as indicated. The following bacteria were grown at 37°C in LB media: *Escherichia coli* strain 1106, *Salmonella typhimurium*, *Enterobacter cloacae β12*, *Providencia rettgeri* strain Dmel, *Providencia burhodogranariea* strain B, *Providencia stuartii* strain DSM 4539, *Providencia sneebia* strain Dmel, *Providencia alcalifaciens* strain Dmel, *Providencia vermicola* strain DSM 17385, *Bacillus subtilis*, and *Staphylococcus aureus*. *Erwinia carotovora carotovora* (*Ecc15*) and *Micrococcus luteus* were grown overnight in LB at 29°C. *Enterococcus faecalis* and *Listeria innocua* were cultured in BHI medium at 37°C. *Candida albicans* was cultured in YPG medium at 37°C. *Aspergillus fumigatus* was grown at room temperature on Malt Agar, and spores were collected in sterile PBS rinses, pelleted by centrifugation, and then resuspended to the desired OD in PBS. The entomopathogenic fungi *Beauveria bassiana* and *Metarhizium anisopliae* were grown on Malt Agar at room temperature until sporulation.

### Systemic infections and survival

Systemic infections were performed by pricking 3- to 5-day-old adult males in the thorax with a 100-μm-thick insect pin dipped into a concentrated pellet of bacteria or fungal spores. Infected flies were subsequently maintained at 25°C for experiments. For infections with *B. bassiana* and *M.*

*anisopliae*, flies were anesthetized and then shaken on a sporulating plate of fungi for 30 s. At least two replicate survival experiments were performed for each infection, with 20–35 flies per vial on standard fly medium without yeast. Survivals were scored twice daily, with additional scoring at sensitive time points. Comparisons of *iso w[1118]* wild-type to *ΔAMPs* mutants were made using a Cox-proportional hazard (CoxPH) model, where independent experiments were included as covariates, and covariates were removed if not significant (p>0.05). Direct comparisons were performed using Log-Rank tests in Prism seven software. The effect size and direction is included as the CoxPH hazard ratio (HR) where relevant, with a positive effect indicating increased susceptibility. CoxPH models were used to test for synergistic contributions of AMPs to survival in R 3.4.4. Total sample size (N) is given for each experiment as indicated.

## Quantification of microbial load

The native *Drosophila* microbiota does not readily grow overnight on LB, allowing for a simple assay to estimate bacterial load. Flies were infected with bacteria at the indicated OD as described, and allowed to recover. At the indicated time post-infection, flies were anesthetized using $CO_2$ and surface sterilized by washing them in 70% ethanol. Ethanol was removed, and then flies were homogenized using a Precellys bead beater at 6500 rpm for 30 s in LB broth, with 300 µl for individual samples, or 500 µl for pools of 5–7 flies. These homogenates were serially diluted and 150 µl was plated on LB agar. Bacterial plates were incubated overnight, and colony-forming units (CFUs) were counted manually. Statistical analyses were performed using One-way ANOVA with Sidak's correction. p-Values are reported as <0.05 = *,<0.01 = **, and <0.001 = ***. For *C. albicans*, BiGGY agar was used instead to select for *Candida* colonies from fly homogenates.

## Gene expression by qPCR

Flies were infected by pricking flies with a needle dipped in a pellet of either *E. coli* or *M. luteus* (OD600 = 200), and frozen at −20°C 6 hr and 24 hr post-infection, respectively. Total RNA was then extracted from pooled samples of five flies each using TRIzol reagent, and re-suspended in MilliQ $dH_2O$. Reverse transcription was performed using 0.5 mg total RNA in 10 µl reactions using Prime-Script RT (TAKARA) with random hexamer and oligo dT primers. Quantitative PCR was performed on a LightCycler 480 (Roche) in 96-well plates using Applied Biosystems SYBR Select Master Mix. Values represent the mean from three replicate experiments. Error bars represent one standard deviation from the mean. Primers used in this study can be found in *Supplementary file 1*. Statistical analyses were performed using one-way ANOVA with Tukey post-hoc comparisons. p-Values are reported as not significant = ns,<0.05 = *,<0.01 = **, and <0.001 = ***. qPCR primers and sources (*Kounatidis et al., 2017*; *Hanson et al., 2016*; *Iatsenko et al., 2016*) are included in *Supplementary file 1*.

## MALDI-TOF peptide analysis

Two methods were used to collect hemolymph from adult flies: in the first method, pools of five adult females were pricked twice in the thorax and once in the abdomen. Wounded flies were then spun down with 15 µl of 0.1% trifluoroacetic acid (TFA) at 21000 RCF at 4°C in a mini-column fitted with a 10 µm pore to prevent contamination by circulating hemocytes. These samples were frozen at −20°C until analysis, and three biological replicates were performed with four technical replicates. In the second method, approximately 20 nl of fresh hemolymph was extracted from individual adult males using a Nanoject, and immediately added to 1 µl of 1% TFA, and the matrix was added after drying. Peptide expression was visualized as described in *Uttenweiler-Joseph et al. (1998)*. Both methods produced similar results, and representative expression profiles are given.

## Melanization and hemocyte characterization, image acquisition

Melanization assays (*Dudzic et al., 2018*) and peanut agglutinin (PNA) clot staining (*Scherfer et al., 2004*) was performed as previously described. In brief, flies or L3 larvae were pricked, and the level of melanization was assessed at the wound site. We used FACS sorting to count circulating hemocytes. For sessile crystal cell visualization, L3 larvae were cooked in $dH_2O$ at 70°C for 20 min, and crystal cells were visualized on a Leica DFC300FX camera using Leica Application Suite and counted manually.

## Acknowledgements

We thank Marc Moniatte and the EPFL proteomics core facility for assistance with MALDI-TOF analysis, Claudia Melcarne for assistance with hemocyte characterization, and Igor Iatsenko for help in preparation of critical reagents. Brian Lazzaro generously provided *Providencia* species used in this study. We thank Hannah Westlake for useful comments on the manuscript. MAH would like to extend special thanks to Jan Dudzic for many illuminating discussions had over coffee.

## Additional information

### Competing interests

Bruno Lemaitre: Reviewing editor, *eLife*. The other authors declare that no competing interests exist.

### Funding

The authors declare that there was no funding for this work

### Author contributions

Mark Austin Hanson, Conceptualization, Data curation, Formal analysis, Validation, Investigation, Methodology, Writing—original draft, Project administration, Writing—review and editing, Designed the study, Performed DrosDel isogenization and recombination, Performed the experiments, Analyzed the data and wrote the manuscript; Anna Dostálová, Conceptualization, Validation, Investigation, Methodology; Camilla Ceroni, Investigation, Provided experimental support; Mickael Poidevin, Shu Kondo, Resources, Methodology, Supplied critical reagents; Bruno Lemaitre, Conceptualization, Resources, Supervision, Funding acquisition, Methodology, Writing—original draft, Project administration, Writing—review and editing, Designed the study, Analyzed the data and wrote the manuscript

### Author ORCIDs

Mark Austin Hanson http://orcid.org/0000-0002-6125-3672
Bruno Lemaitre http://orcid.org/0000-0001-7970-1667

### Decision letter and Author response

Decision letter https://doi.org/10.7554/eLife.44341.019
Author response https://doi.org/10.7554/eLife.44341.020

## Additional files

### Supplementary files

• Supplementary file 1. Primers used in this study.
DOI: https://doi.org/10.7554/eLife.44341.016

• Transparent reporting form
DOI: https://doi.org/10.7554/eLife.44341.017

### Data availability

Data generated or analysed during this study are included in the manuscript and supporting files. Source data has been provided for Figure 2.

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
