## [Decision Letter]

Thank you for submitting your article "Synergy and remarkable specificity of antimicrobial peptides in vivo using a systematic knockout approach" for consideration by *eLife*. Your article has been reviewed by three peer reviewers, and the evaluation has been overseen by a Reviewing Editor and Wendy Garrett as the Senior Editor. The following individuals involved in review of your submission have agreed to reveal their identity: Mathias Hornef (Reviewer #2); Lora V Hooper (Reviewer #3).

The reviewers have discussed the reviews with one another and the Reviewing Editor has drafted this decision to help you prepare a revised submission.

It has been agreed that the work is of a high standard and that further experimentation is not required. Please address the points of interpretation and the necessary corrections that are included in the critiques below. Please also consider the suggestions of the reviewers for improvements in the manuscript.

*Reviewer #1:*

In the first large-scale loss-of-function study of antimicrobial peptide (AMP) genes in *Drosophila melanogaster*, the authors demonstrate a marked difference in the role of AMPs in the two innate immune pathways. In the Imd pathway, AMPs are essential. Moreover, a single AMP (DiptA) is necessary for defense against a particular bacterium (*P. rettgeri*). In the Toll pathway, AMPs have a secondary role to the Bomanin peptides. None of the AMPs, individually or collectively, is necessary for survival against Gram positive bacteria and fungi. The experiments are carried out to high standards, thorough, and worthy of publication in this journal. Nevertheless, there are several problems in presentation and interpretation:

The term "synergy" is applied incorrectly.

Certain AMPs, such as PGLa and magainin-2, act synergistically: each alone is modestly effective but the two together have a potent antimicrobial activity. Here, there is no evidence for synergy between AMPs. The assay is for survival, which is not a quantitative measure of activity. Consider two fly AMPs. In scenario one, flies missing the gene for either AMP have 100% survival, but the double mutant has 0% survival. In scenario two, both single mutants and the double have 0% survival. Although in scenario one the phenotypic effect of combining mutations is not additive, the AMPs could be fully redundant, with less than 50% activity required for full survival. Furthermore, in scenario two, the AMPs could have strong biochemical synergy, with no antimicrobial activity and hence no survival unless the both gene products are present. Given the well-established biochemical definition of synergistic action for certain pairs of AMPs and the data here indicating redundancy, not synergy, among the *Drosophila* AMPs, the authors should not use "synergy" and "synergistic" to describe their findings.

Previously, the strongest genetic information on AMP function in *Drosophila* immunity was an elegant 2002 study from the same lab (Tzou et al.), where they studied the effect on survival of heterologous expression of single AMP genes when both the Imd and Toll pathways were inactivated. They found that Defensin was protective against *B. subtilis*, Attacin A against Ecc15, and Drosomycin protective against *A. fumigatus*. However, here they find that eliminating Defensin, Attacins, Drosomycin, and all other AMPs has only a slight effect on resistance to two of these three pathogens. The authors are encouraged to point out and discuss these differences.

The authors report a "curious" fact: the effect on survival of eliminating a relevant set of AMP genes is ameliorated by eliminating a seemingly irrelevant set of AMP genes. They offer three explanations, but overlook a fourth that seems probable. Pathogens and hosts share many molecular and cellular features. Specificity against "microbes" is therefore likely to be limited. Furthermore, AMPs are required at quite high concentrations. It could easily be, therefore, that AMPs in general, and certain AMPs in particular, can be toxic to their hosts by, for example, weakening host cell membranes. Wild-type host cells could survive, but host cells with reduced disease resistance might be killed. If so, deleting the genes for such AMPs could reduce toxicity in the context of a crippled immune system without invoking ineffective AMPs, interactions among AMPs, or genetic background.

*Reviewer #2:*

Hanson et al. describe in their manuscript entitled “Synergy and remarkable specificity of antimicrobial peptides in vivo using a systematic knockout approach" the generation of *D. melanogaster* devoid of all (14 individual ones), groups thereof or individual antimicrobial peptides (AMP) and their functional characterization in appropriate infection models with a range of bacterial and fungal microorganisms. They compared and assigned the phenotype of AMP deficient mutants to major signaling pathways (imd and toll) and the newly described group of Bomanins. They detected an additive/synergistic effect of individual peptides and a surprising specificity of some AMPs against some bacterial organisms. The study represents a significant contribution to the field answering a long-standing question namely the evolutionary benefit of the many AMPs for antimicrobial host defense in D.m.. The work is extensive and paves the way to a future detailed understanding of the evolutionary driving forces and disease susceptibility factors and has exciting implications for other model systems (e.g. other insects, mice) and humans. The topic is timely and exciting, the technical approaches are careful and sound (e.g. Figure 1B, Figure 1—figure supplement 2), elegant (Figure 5—figure supplement 1) and overall state of the art. The work appears suitable for publications in *eLife*.

1) In addition to the possible explanations listed in the subsection “Drosomycin and Metchnikowin additively contribute to defence against the yeast *C. albicans*”, the lack of individual/groups of AMPs may lead to the compensatory increase in the expression of other AMPs. A careful transcriptional analysis of at least certain representative AMPs from the same group/the other groups might strengthen the study.

2) AMPs have also been implicated in the homeostatic host-microbial interaction. Although the reviewer feels that an in depth analysis of the complex homeostatic interaction is beyond the scope of the present work, a short description of major alterations (e.g. lifespan, general level of inflammation, endogenous infection) of the complete AMP (14-fold) ko would be interesting and stimulating.

3) The effect of AMPs in vivo may -in addition to their antimicrobial spectrum- depend on their anatomical distribution (fat body vs. peripheral), kinetic of expression and overall expression magnitude. These aspects are mentioned but should be discussed in the context of the specificity/general activity of peptides.

*Reviewer #3:*

This paper has accomplished the ambitious goal of characterizing the in vivo function of each known antimicrobial peptide (AMP) encoded in the *Drosophila* genome. For technical reasons this was not possible before the advent of CRISPR-based genetics, and the authors have now harnessed CRISPR technology to create mutations in each of the individual AMP genes as well as to create mutants that lack all AMPs, or subsets of AMPs. They have performed extensive phenotypic characterization of these AMP-deficient *Drosophila* and have determined the response of each to infection with a battery of microorganisms. Their findings pinpoint for the first time the physiological functions of individual AMPs, as well as identifying synergies among AMP groups.

Altogether, this is an important study that comprehensively illuminates the physiological functions of *Drosophila* AMPs. The study is thorough, the findings convincing, and the results well-presented and clear. There is little to complain about here, and so I have just a few comments regarding the clarity of the figures and the Discussion.

1) It would be helpful if the authors could include, either as part of one of the figures or as a separate display item, a table that lists the various mutants that they analyzed and which AMPs or groups of AMPs are targeted in each mutant. This will make it much easier to follow along with the phenotypic data in each of the figures.

2) The authors have a good discussion of how their data helps to explain why there is such a diversity of AMPs. Is it also possible that different AMPs defend different spatial or temporal niches and would this be worth also including in the Discussion?

3) It seems like it would also be worth discussing whether some of the AMPs also determine the composition of resident microbial communities in *Drosophila*. The authors did not analyze for this and I'm not suggesting that they do so given the already broad scope of the paper, but it might be a good point to bring up in the Discussion.

---

## [Author Response]

It has been agreed that the work is of a high standard and that further experimentation is not required. Please address the points of interpretation and the necessary corrections that are included in the critiques below. Please also consider the suggestions of the reviewers for improvements in the manuscript.

Reviewer #1:

[…] The term "synergy" is applied incorrectly.Certain AMPs, such as PGLa and magainin-2, act synergistically: each alone is modestly effective but the two together have a potent antimicrobial activity. Here, there is no evidence for synergy between AMPs. The assay is for survival, which is not a quantitative measure of activity. Consider two fly AMPs. In scenario one, flies missing the gene for either AMP have 100% survival, but the double mutant has 0% survival. In scenario two, both single mutants and the double have 0% survival. Although in scenario one the phenotypic effect of combining mutations is not additive, the AMPs could be fully redundant, with less than 50% activity required for full survival. Furthermore, in scenario two, the AMPs could have strong biochemical synergy, with no antimicrobial activity and hence no survival unless the both gene products are present. Given the well-established biochemical definition of synergistic action for certain pairs of AMPs and the data here indicating redundancy, not synergy, among the Drosophila AMPs, the authors should not use "synergy" and "synergistic" to describe their findings.

We thank reviewer 1 for their kind assessment. Reviewer 1 is correct here in that synergy has a well-defined implication which we do not demonstrate through survival data and bacterial load assays. We intended for the use of “synergy” here to reflect statistical interactions, not an implication of the underlying mechanism; see subsection “AMPs act additively and synergistically to suppress bacterial growth in vivo” of the revised manuscript that may help address this concern. We appreciate the comment and have revised our language to avoid misuse of this term (see revised manuscript, e.g. Abstract and subsection “AMPs synergistically contribute to defence against P*. burhodogranariea*”, second paragraph).

Previously, the strongest genetic information on AMP function in Drosophila immunity was an elegant 2002 study from the same lab (Tzou et al.), where they studied the effect on survival of heterologous expression of single AMP genes when both the Imd and Toll pathways were inactivated. They found that Defensin was protective against B. subtilis, Attacin A against Ecc15, and Drosomycin protetive against A. fumigatus. However, here they find that eliminating Defensin, Attacins, Drosomycin, and all other AMPs has only a slight effect on resistance to two of these three pathogens. The authors are encouraged to point out and discuss these differences.

We now discuss Tzou et al., 2002, in the revised manuscript for these observations in our Discussion; see subsection “A combinatory approach to study AMPs”.

The authors report a "curious" fact: the effect on survival of eliminating a relevant set of AMP genes is ameliorated by eliminating a seemingly irrelevant set of AMP genes. They offer three explanations, but overlook a fourth that seems probable. Pathogens and hosts share many molecular and cellular features. Specificity against "microbes" is therefore likely to be limited. Furthermore, AMPs are required at quite high concentrations. It could easily be, therefore, that AMPs in general, and certain AMPs in particular, can be toxic to their hosts by, for example, weakening host cell membranes. Wild-type host cells could survive, but host cells with reduced disease resistance might be killed. If so, deleting the genes for such AMPs could reduce toxicity in the context of a crippled immune system without invoking ineffective AMPs, interactions among AMPs, or genetic background.

We thank reviewer 1 for the suggestion, and have included in the revised version this idea into the “complex biochemical interactions amongst the AMPs involved,” hypothesis; see subsection “Drosomycin and Metchnikowin additively contribute to defence against the yeast *C. albicans*”.

Reviewer #2:

[…] 1) In addition to the possible explanations listed in the subsection “Drosomycin and Metchnikowin additively contribute to defence against the yeast C. albicans”, the lack of individual/groups of AMPs may lead to the compensatory increase in the expression of other AMPs. A careful transcriptional analysis of at least certain representative AMPs from the same group/the other groups might strengthen the study.

We thank reviewer 2 for their kind words. This hypothesis intrigued us as well, however we feel this requires careful consideration of confounding effects (timespan of maximal expression, IMs produced, biochemistry of IMs and bacteria in the absence of AMPs), and is beyond the scope of this study.

2) AMPs have also been implicated in the homeostatic host-microbial interaction. Although the reviewer feels that an in depth analysis of the complex homeostatic interaction is beyond the scope of the present work, a short description of major alterations (e.g. lifespan, general level of inflammation, endogenous infection) of the complete AMP (14-fold) ko would be interesting and stimulating.

We appreciate that this topic is of great interest. We did not see drastic differences in lifespan between our iso wild-type and the δ-AMPs flies but the situation is quite complex as sub-groups of AMPs seems to impact life span. For instance, we have found that combined chromosome 2 AMP mutants have longer lifespan; this is an arbitrary genotype combination that we do not include in the current manuscript. While it would be possible to add this data to the revised manuscript, this may generate confusion and distract from the immunity message. Additionally, we feel these lifespan data need to be repeated and the interpretation of these data is far more complex and would likely require additional experiments, both of which would delay the paper. Therefore we would prefer not to include these data. In our concluding remarks (subsection “On the role of AMPs in host defence”, last paragraph), we state that the AMP-deficient flies can be used to address the role of AMPs beyond infection.

3) The effect of AMPs in vivo may -in addition to their antimicrobial spectrum- depend on their anatomical distribution (fat body vs. peripheral), kinetic of expression and overall expression magnitude. These aspects are mentioned but should be discussed in the context of the specificity/general activity of peptides.

While these tissue-specific effects are of great importance to natural modes of infection, our major model of systemic infection bypasses surface epithelia focusing solely on the systemic immune response, relying overwhelmingly on AMP production by the fat body and also hemocytes, but not other tissues. We will include this consideration and response in the Discussion to clearly state that further studies need to analyse the role of AMP in epithelia immunity, notably the gut, genitalia, and malpighian tubules. See subsection “A combinatory approach to study AMPs”, and subsection “On the role of AMPs in host defence”.

Reviewer #3:

[…] Altogether, this is an important study that comprehensively illuminates the physiological functions of Drosophila AMPs. The study is thorough, the findings convincing, and the results well-presented and clear. There is little to complain about here, and so I have just a few comments regarding the clarity of the figures and the Discussion.1) It would be helpful if the authors could include, either as part of one of the figures or as a separate display item, a table that lists the various mutants that they analyzed and which AMPs or groups of AMPs are targeted in each mutant. This will make it much easier to follow along with the phenotypic data in each of the figures.

We are happy to further describe the nature of each mutation in Figure 1—figure supplement 1, including which combinatory group these mutations belonged to, and thank the reviewer for this suggestion to make our data more accessible. See subsection “Generation and characterization of AMP mutants”.

2) The authors have a good discussion of how their data helps to explain why there is such a diversity of AMPs. Is it also possible that different AMPs defend different spatial or temporal niches and would this be worth also including in the Discussion?

This is an intriguing proposal. Lemaitre et al., 1997, describe the kinetics of AMP induction, noting that different AMPs begin to be expressed at slightly varying time points. However we do not think we can incorporate this into our Discussion satisfyingly with the data we have presented as we mainly use systemic infection, rapidly inducing AMP production by the fat body and hemocytes. As stated above in our response to reviewer 2, we have mentioned that the role of AMPs in epithelial immunity remains to be investigated.

3) It seems like it would also be worth discussing whether some of the AMPs also determine the composition of resident microbial communities in Drosophila. The authors did not analyze for this and I'm not suggesting that they do so given the already broad scope of the paper, but it might be a good point to bring up in the Discussion.

The AMP-microbiota interaction will surely be of interest for future studies. We acknowledge this area of research in opening (Introduction, second paragraph) and closing (subsection “On the role of AMPs in host defence”, second paragraph) comments. As we do not closely investigate microbiota interactions, we feel this is best left for future studies to discuss and elucidate.